A novel framework for three-dimensional electrical impedance tomography reconstruction of maize ear via feature reconfiguration and residual networks

http://orcid.org/0000-0001-5014-4799 Zheng Hai-Ying 1 2
Li Yang 1 3
Wang Nan 1 2
Xiang Yang 1 3
Liu Jin-Hang 1 2
Zhang Liu-Deng 1 3
Huang Lan 1 2 hlan@cau.edu.cn
Wang Zhong-Yi 1 3
1 College of Information and Electrical Engineering, China Agricultural University , Beijing , China
2 Key Laboratory of Agricultural Information Acquisition Technology (Beijing), Ministry of Agriculture , Beijing , China
3 Key Laboratory of Modern Precision Agriculture System Integration Research, Ministry of Education , Beijing , China
Angiulli Giovanni
Electronic publication date: 2024 Apr 11
Publication date: 2024
Volume: 10
Electronic Location ID: e1944
Received 2023 Oct 3; Accepted 2024 Feb 27
Copyright: © 2024 Zheng et al.
Copyright year: 2024
Copyright holder: Zheng et al.
License: This is an open access article distributed under the terms of the Creative Commons Attribution License, which permits unrestricted use, distribution, reproduction and adaptation in any medium and for any purpose provided that it is properly attributed. For attribution, the original author(s), title, publication source (PeerJ Computer Science) and either DOI or URL of the article must be cited.
License URL: https://creativecommons.org/licenses/by/4.0/

Keywords: Electrical impedance tomography (EIT), Maize ear, 3D conductivity distribution, Feature reconfiguration, Residual networks

Funding: National Natural Science Foundation of China 32171891 and 62271488 This work was supported by the National Natural Science Foundation of China under Grants 32171891 and 62271488. The funders had no role in study design, data collection and analysis, decision to publish, or preparation of the manuscript.

==============================
Electrical impedance tomography (EIT) provides an indirect measure of the physiological state and growth of the maize ear by reconstructing the distribution of electrical impedance. However, the two-dimensional (2D) EIT within the electrode plane finds it challenging to comprehensively represent the spatial distribution of conductivity of the intact maize ear, including the husk, kernels, and cob. Therefore, an effective method for 3D conductivity reconstruction is necessary. In practical applications, fluctuations in the contact impedance of the maize ear occur, particularly with the increase in the number of grids and computational workload during the reconstruction of 3D spatial conductivity. These fluctuations may accentuate the ill-conditioning and nonlinearity of the EIT. To address these challenges, we introduce RFNetEIT, a novel computational framework specifically tailored for the absolute imaging of the three-dimensional electrical impedance of maize ear. This strategy transforms the reconstruction of 3D electrical conductivity into a regression process. Initially, a feature map is extracted from measured boundary voltage via a data reconstruction module, thereby enhancing the correlation among different dimensions. Subsequently, a nonlinear mapping model of the 3D spatial distribution of the boundary voltage and conductivity is established, utilizing the residual network. The performance of the proposed framework is assessed through numerical simulation experiments, acrylic model experiments, and maize ear experiments. Our experimental results indicate that our method yields superior reconstruction performance in terms of root-mean-square error (RMSE), correlation coefficient (CC), structural similarity index (SSIM), and inverse problem-solving time (IPST). Furthermore, the reconstruction experiments on maize ears demonstrate that the method can effectively reconstruct the 3D conductivity distribution.

Introduction

During the growth of maize, a series of physical, physiological, and biochemical changes occur, and then the spatial distribution of the moisture content of an intact maize ear serves as a crucial determinant of the physiological status and growth of the plant (Clark et al., 1997; Wang et al., 2019; Nilahyane et al., 2020). Fortunately, the electrical impedance spectrum of the maize ear has a favorable correlation with its moisture content; thus, the change in its moisture content can be reflected by the electrical impedance distribution (Zhao et al., 2015; Li et al., 2022). Moreover, electrical impedance tomography (EIT), a non-invasive, non-radiometric imaging modality, can reconstruct the conductivity distribution within a field by recording the voltage measured at the domain boundary (Barber & Brown, 1984; Newell, Gisser & Isaacson, 1988; Yao & Takei, 2017; Ke et al., 2022). Thus, EIT allows us to obtain spatial moisture states through conductivity distribution. In fact, EIT has been widely applied in many fields. In the medical field, it has been employed for pulmonary function screening (Martins et al., 2019; Shi et al., 2021; Fossali et al., 2022), functional brain imaging (Jiang & Soleimani, 2019; Ke et al., 2022), and cancer detection (Wu & Soleimani, 2019; Sun et al., 2019; Mansouri et al., 2022). Industrial applications include material damage detection (Mao et al., 2019), artificial skin (Liu et al., 2020), and industrial simulation (Wang, Giorgio-Serchi & Yang, 2022). In the field of agriculture, EIT has been harnessed for observing plant growth (Corona-Lopez et al., 2019; Ehosioke et al., 2020), root development (Wang et al., 2023), and resistance assessment (Qian et al., 2021), among other uses.

Based on electromagnetic field theory, EIT indicates that the distribution and propagation trajectory of the electric field depend on the material dispersion in the imaging region, resulting in a three-dimensional (3D) diffusion within the field (Liu et al., 2019). This essentially constitutes a 3D vector field. In measurements of maize ears, the excitation current exhibits omnidirectional diffusion through the husk, kernels, and cob into the 3D field. The spatial conductivity distribution within this 3D domain can reflect the growth across various components of the maize ear. However, there is a limitation to two-dimensional (2D) EIT. It only facilitates the reconstruction of the conductivity distribution on the plane occupied by the electrode, based on the boundary voltage. This method is difficult for obtaining longitudinal information within the maize ear; specifically, “longitudinal” represents the Z-axis, a measure of the Z-axis position (perpendicular to the observational plane) in 3D space. In contrast, 3D EIT enables the acquisition of comprehensive data on the conductivity distribution of the maize ear. This data indirectly illuminates the current physiological state of the maize ear.

Presently, research on 3D EIT reconstruction algorithms is largely centered on traditional numerical computation-based methods and interpolation reconstruction methods. It is noteworthy that most of these methods employ differential imaging to mitigate the noise interference. Metherall et al. (1996), pioneers in this field, recognized early on that 2D EIT reconstruction would significantly impede the imaging capabilities of EIT. They integrated a 64-electrode data collection device with a matrix inversion technique to propose a 3D EIT system. Given that the electric current diffuses through a three-dimensional space, Vauhkonen et al. (1999) argued that the non-electrode planar structures would have a significant impact on the reconstruction results. Consequently, they proposed a methodology for solving the finite element approximation and the generalized Tikhonov regularized inverse problem based on a 3D EIT all-electrode model. Zhang et al. (2019) proposed an iterative reconstruction algorithm based on the Gauss-Newton method. This algorithm applies a multiplicative regularization scheme to the 3D EIT inverse problem in order to overcome its ill-conditioned nature. They managed to reconstruct reasonable lung anatomy using differential imaging, but the challenge of absolute imaging remains unresolved. Basak & Wahid (2022) developed an in-situ measurement system for the root zone, applicable for sensor-based monitoring of root growth via differential 3D imaging. However, these 3D reconstruction methods necessitate substantial computational resources to solve the inverse problem. Alternatively, Wang et al. (2023) employed 2D slices to reconstruct 3D images, thereby reducing computational efforts. They proposed using weighted frequency difference imaging to effectively suppress background noise in reconstructed images. Moreover, they observed irregular shapes in the root zone distribution of the plant species peony. Li et al. (2021) utilized a linear interpolation method to enhance the imaging speed of lung EIT images. In contrast, Shin, Ahmad & Mueller (2021) achieved superior spatial resolution in the vertical direction (Z-axis) by stacking multilayer two-dimensional slices based on the Calderón method. Li et al. (2022) first employed the EIT technique for moisture content measurement in maize ear and successfully visualized moisture content distribution in the 3D maize ear via interpolation reconstruction of multilayer conductivity. Nevertheless, despite the improved imaging speed and 3D structure visualization offered by interpolation or stacked slicing methods, they can only represent the 2D conductivity distribution in the plane of the electrode layers. This approach inherently is difficult to capture key structural details, specifically the unreconstructible continuous conductivity distribution in the longitudinal (Z-axis) direction. Additionally, differential imaging is unable to recreate the observed object’s actual conductivity value. It is so challenging for these interpolation-based 3D visualization techniques to accurately depict the continuous and whole conductivity distribution of the maize ear.

There is a soft field effect in EIT imaging; that is, the sensitivity distribution of the sensitive field is not uniform and is affected by the global distribution of the measured medium. This leads to a nonlinear correspondence between the boundary voltage and the internal conductivity distribution of the maize ear. Additionally, the weak coupling characteristic of the maize ear results in a change in the contact impedance between the excitation electrode and the husk, as well as the presence of voids inside the plant body. These factors contribute to an increase in noise, exacerbating the ill-conditioned problem of EIT. During the growth period of the maize ear, the moisture content of the husk, kernels, and cob fluctuates widely. Thus, the true conductivity distribution needs to be quantitatively described by absolute imaging. As is well known, absolute imaging is sensitive to noise, making it difficult to obtain stable imaging results. Therefore, achieving excellent 3D imaging of the maize ear remains a challenge.

With the development of deep learning, convolutional neural networks (CNNs) possess the capability to autonomously learn complex, nonlinear function mappings from large-scale datasets. Importantly, they also offer the flexibility of incorporating noise as prior information into the model. At present, CNNs have emerged as a representative method for deep learning in 2D EIT. For instance, Tan et al. (2019) used a CNN-based approach for EIT tasks, and their experimental results demonstrated superiority over traditional algorithms. Li et al. (2020) proposed a one-dimensional CNN algorithm that achieves better edge shape reconstruction. Similarly, Wu et al. (2021) modeled a nonlinear mapping between measured voltage and conductivity distribution using CNN. The class of methods can improve the feature extraction capability of the network by increasing the number of neural network layers, also known as stacking layers, so that the model can learn and capture more complex features and patterns, which helps the network to better fit complex data. However, with the increase in the number of layers, deep networks are prone to the problem of gradient vanishing or gradient explosion, which leads to training difficulties and triggers network degradation (Srivastava, Greff & Schmidhuber, 2015; He et al., 2016; He & Sun, 2015). In addition to CNNs, another class of mainstream methods is based on image segmentation ideas. For example, Wang et al. (2022a) proposed the ResV-Net method to tackle the nonlinear, ill-posed inverse problem, and both simulations and physical experiments have shown that this method yields improved visualization results. Zhang et al. (2022) introduced a V-shaped dense denoising Net (VDD-Net), based on CNN, to express the nonlinear mapping between the measurements and the parameters in the observation domain. Subsequently, Wang et al. (2022b) proposed V2D−Net, a high-resolution reconstruction algorithm containing a regularized reconstruction module and a multichannel convolutional network, whose effectiveness has been experimentally demonstrated. Moreover, Ye et al. (2023) extended image segmentation methods to 3D EIT reconstruction, using U2-Net to solve the 3D EIT inverse problem. While simulation results indicate that this method significantly improves the quality of 3D reconstructions, it has not yet been validated through physical experiments. Inspired by the 3D-GAN model, Yi, Chen & Yang (2022) introduced a transposed convolutional neural network (TN-Net) for 3D image reconstruction; however, GANs are susceptible to challenges like mode collapse and training instability. Yang’s team has contributed methods extendable to 3D imaging for monitoring cell culture processes, such as DL-GS (Chen et al., 2021), SADB-Net (Chen & Yang, 2021), and MSFCF-Net (Liu, Bagnaninchi & Yang, 2022). These techniques enable imaging on physical models with high signal-to-noise ratios (SNR) and small scales. Despite the promise of deep learning in EIT absolute imaging as a fast, nonlinear method for solving inverse problems, several challenges persist. EIT is sensitive to electrode properties, and the effect of contact impedance must be considered during image reconstruction (Vilhunen et al., 2002; Boyle & Adler, 2011). In maize ear measurement experiments, the contact impedance between the electrode array and the husk changes (Li et al., 2022). Furthermore, in contrast to simulation experiments, the noise present in actual measurement data is more complex. All these factors can lead to unstable neural network training, an increased risk of overfitting, and a dependency on data augmentation. Neural network models are highly susceptible to excessively memorizing noise and details in the training data during training, neglecting the overall trends and generalization capabilities of the data. This leads to good performance on the training data but poor performance on test data, causing overfitting (Algan & Ulusoy, 2021). Consequently, the application of deep learning-based 3D EIT absolute imaging for maize ears continues to face formidable challenges.

Existing studies on the 3D conductivity reconstruction of the maize ear, such as our previous work by Li et al. (2022), has tried to visualize the 3D conductivity distribution using a 2D planar interpolation method. However, this approach does not extend to reconstructing the comprehensive and continuous conductivity distribution in the Z-axis direction. The inverse problem-solving mechanisms of absolute imaging methods, epitomized by the Gauss-Newton (GN) approach, suffer from limited efficiency. These methods face significant challenges in reconstructing the conductivity distribution of targets characterized by multi-layered structures, such as maize ears. Specifically, differentiating the conductivities of the husk, kernels, and cob presents a formidable task. In order to solve the problems associated with the absolute reconstruction of 3D conductivity in maize ears, the reconstruction process is understood from the perspective of regression analysis. This enables us to learn the nonlinear mapping model of the unknown physical system from the data, thereby completing 3D EIT absolute imaging. Considering that 3D EIT necessitates the reconstruction of a larger amount of conductivity data, directly stacking CNNs to enhance reconstruction quality could lead to network degradation and instabilities during the training process. Consequently, our method builds upon the deep residual network (ResNet) architecture (He et al., 2016). The main contributions of this article are as follows:

(1) This research presents a single-channel feature reconstruction network framework specifically designed for absolute imaging in three-dimensional electrical impedance tomography (3D EIT). The framework addresses the ill-conditioning and nonlinearity in EIT by transforming 3D conductivity reconstruction into a regression process.

(2) The severe ill-posedness problem poses a significant challenge for 3D EIT reconstruction, mainly due to the limitations of limited voltage data and fixed excitation. At the initial stage, we use the data reconstruction module to extract the feature map from the measured voltage signals. This not only enhances the correlation between different dimensions but also facilitates the extraction of more correlation features between the data, thereby alleviating the ill-posedness present in the EIT reconstruction process.

(3) The applicability and effectiveness of RFNetEIT have been validated through numerical simulation experiments, acrylic model experiments, and fresh maize ear experiments. The experimental results show that, compared to existing methods, this approach has significant advantages in terms of root-mean-square error (RMSE), correlation coefficient (CC), structural similarity index (SSIM), inverse problem-solving time (IPST), as well as in 3D visualization effects.

The remainder of the article is organized as follows: “Methods” introduces the 3D EIT reconstruction algorithm, while “Materials and Datasets” presents the hardware and data used in the experimental studies. “Results” provides the real measurements and simulation results, followed by a discussion of these findings in “Discussion”. Finally, “Conclusions” offers the conclusions of the research.

Methods

Forward and inverse problems in 3D EIT

The two main elements that define EIT are the forward problem and the inverse problem. In the forward problem, the potential distribution ϕ across the field is calculated using the previously determined excitation current I→ and conductivity distribution σ, establishing the framework for the inverse problem and empirical measurement. The challenges in 3D electrical impedance imaging include a broad solution field range, complex geometry, and more. By constructing the complete electrode model (CEM) and finite element method (FEM), the three-dimensional measurement domain Ω is discretized into N tetrahedral units. This turns a continuous, infinite degree of freedom problem into a discrete, finite degree of freedom problem; that is, the continuously varying conductivity σ0 in the measurement domain is discretized into N elements, as shown in Eq. (1):

(1) σΩ→discretizeσℝ={σ1,σ2,…,σN−1,σN}Ω.

Solving the forward problem by Eq. (2):

(2) ϕ(x,y,z)=U(σR,I→),(x,y,z)∈Ω

where U(σR,I→) signifies the mapping relationship used to solve for the field potential ϕ, using the excitation current vector I→ and the discrete conductivity σR.

Equations (3) and (4) concisely summarize the inverse problem, which aims to solve for the internal electrical impedance distribution σR, given the known excitation current I→ and boundary measurement voltage ϕL.

(3) Δϕ(x,y,z)=JΔσR

(4) σR′(x,y,z)=F(ϕl−ε),l∈[1,L].

The amount of change ΔσR in the conductivity distribution σR is what causes the boundary voltage perturbation, which is represented by the letter Δϕ(x,y,z). F(ϕl−ε) represents the imaging methodology, J stands for the Jacobian matrix, also known as the sensitivity matrix, and ϕl represents the boundary measurement voltage at a particular place. The hardware system adopts an adjacent excitation and measurement pattern, sequentially exciting and measuring in pairs. Figure 1 shows the first three excitations. Initially, electrodes 2 and 1 are excited, and the measured electrode pairs include [4,3], [5,4],… [48,47], [33,48]. A measurement cycle consists of 45 electrode pairs, excluding the excited pair. Then, electrodes 3 and 2 are excited to complete another measurement cycle; this is followed by the excitation of electrodes 4 and 3. The process continues until electrodes 1 and 16 are excited, resulting in 16 excitations for the first layer of electrodes. All three layers of electrodes use the same excitation method, totaling 48 excitations and 2,160 measurements. L = 2,160. System noise is represented by ε.

Figure 1 Excitation and measurement pattern for hardware systems.

The three images in the first row represent three layers of electrode rings. The first stimulation occurs on electrodes No. 2 and No. 1 of the A electrode ring, while the remaining electrodes of the A ring and all electrodes of the B and C rings are used for measurement, with the measurement sequence following a counterclockwise rotation. The images in the second row indicate the second stimulation, with electrodes No. 3 and No. 2 being stimulated, and the rest of the setup is the same as in the first row. Similarly, the third row represents the third stimulation occurring on electrodes No. 4 and No. 5.

Reconstruction of conductivity based on deep learning

The mathematical expression for EIT image reconstruction is depicted in Eq. (4). When given a known ϕL, t the endeavor to solve for the discrete variable σR to approximate σΩ presents an ill-conditioned nonlinear inverse problem. CNNs, as data-driven, general-purpose, function-based approximators, utilize nonlinear activation functions at every network layer. This allows the network to encapsulate nonlinear relationships. Furthermore, the design of convolutional layers, characterized by a local receptive field and a weight-sharing architecture, ensures each neuron connects only to a small portion of the input data. Moreover, neurons within the same layer share identical weights. This structure equips CNNs with the capability to discern local and translation-invariant features in high-dimensional data, such as images. As a result, they are adept at managing nonlinear and intricate data. Utilizing their deep architecture and convolutional operations, CNNs can autonomously learn and distill salient features from data. In an effort to minimize predictive errors, these networks employ techniques like back-propagation and gradient descent for parameter optimization. Importantly, during the training process, the multilayered architecture of CNNs facilitates a seamless automated feature extraction pathway. This pathway transitions from raw data to low-level features and then to high-level composite features. Notably, this automated process significantly reduces the condition number required for computational solutions, thereby enhancing efficiency. Such pronounced efficiency is particularly effective in addressing challenges associated with ill-conditioned scenarios.

Additionally, regularization tactics, such as weight decay, dropout, and batch normalization, when implemented within CNNs, are instrumental in addressing ill-conditioned problems and simultaneously enhancing the model’s generalization proficiency. Equation (5) demonstrates the CNN approach to tackle this problem. Here, F represents the network topology, E and g denote the cost function and regularization term, respectively, and θ symbolizes the network parameter. This approach eliminates the necessity of calculating the sensitivity matrix J or manually determining components like the forward model, cost function, regularization, and optimizer (Aggarwal, Mani & Jacob, 2018; Ren et al., 2020; Zhang et al., 2022; Chen, Yang & Bagnaninchi, 2021). Deep learning-based inverse problem-solving frameworks effectively tap into the correlation between potential distribution information in the spatial domain, thus producing an optimal model with formidable generalization abilities within a recognized data spectrum.

(5) FLearned=argmin∑m=1M⁡(E{σ,Fθ(ϕ)})+g(θ).

In contrast to the two-dimensional EIT reconstruction task, σ(x,y,z) encompasses not only the conductivity information σ(x,y) in the electrode layer plane but also integrates the conductivity information σ(x,y)(z) from the non-measurement plane. This is elucidated in Eq. (6), where z signifies the position of the measurement plane along the vertical axis. Drawing inspiration from Eq. (6), we adopt the representation of conductivity as depicted in Eq. (7) for this study.

(6) σΩ=∮Ω⁡σ(x,y)(z)dz

(7) σΩ=∰Ωσ(x,y,z)dxdydz.

Following the fusion of Eqs. (1) and (7), Eq. (8) provides the mathematical representation of the approximate conductivity set after field discretization. N represents the number of cells after the finite element method (FEM) computation.

(8) {∰Ωσ(x,y,z)dxdydz≈∑n=1N{σ1,σ2,…,σN−1,σN=95295}{σ1,σ2,…,σN−1,σN=95295}⊆Ω.

Single channel feature reconfiguration network for 3D EIT

In EIT imaging, a sensitivity matrix typically characterizes the relationship between the boundary-measured voltage changes and the internal electrical conductivity variations. Given that the potential distribution is not limited to a 2D plane, data collected from the boundary of the measurement domain presents challenges in reconstructing a comprehensive 3D image. In this work, an end-to-end approach is used to couple arbitrary nonlinear functions Flearned:ϕL→σR, thereby circumventing the need for sensitivity matrix computation. In fact, differences in tissue electric properties of measurement domain have an effect on distribution of electric field, leading to varied boundary voltage changes. While image reconstruction premised on minor boundary voltage alterations may be efficacious in 2D EIT reconstruction grounded in image segmentation theory, this method underperforms in 3D EIT, which is characterized by a larger number of reconstruction blind spots.

Therefore, we address the 3D EIT reconstruction problem from the perspective of data distribution, taking into account the subtle differences in boundary voltage. We then opt for residual mapping instead of the direct mapping typically used in traditional CNN approaches.

Principle of feature reconstruction

Based on the matrix design guidelines presented in Eq. (9), the measurements of the boundary voltage Vb include two types of information: (1) univariate data derived from identical excitation electrodes but varying measurement electrodes (information across matrix rows), and (2) bivariate data derived from distinct excitation and measurement electrodes (information across matrix columns). In this study, we employ single-channel convolution to extract the amalgamated features, which are represented in the two types of information embedded in the reconstructed feature map, as illustrated in Fig. 2D. Unlike the one-dimensional convolution strategy employed by Li et al. (2020), our method places greater emphasis on the feature differences across distinct measurement objects. Additionally, sections c and d of Fig. 2 highlight the unique features derived from both column and row matrices, which themselves originate from variations in the excitation and measurement electrodes. Given the pronounced ill-posedness characteristic of 3D EIT, our method extracts features that are more differentiated compared to the one-dimensional sequence shown in Fig. 2A. Furthermore, the implicit information contained in the original data is also preserved to the greatest extent possible, avoiding the possibility of correlation patterns and data distribution being altered as a result of forced transformations from single-channel one-dimensional data to multi-channel two-dimensional data (Zhu et al., 2021). In our experiments, using Vb for projection, and the feature map Fig. 2C is transposed according to Fig. 2D. The main purpose is to adapt to the computational rules of the convolutional sliding window, and to prioritise the computation of the row data with the most obvious feature differences. This data representation is more amenable to the “end-to-end” processing mode of CNNs compared to the original voltage sequences.

Figure 2 Two-dimensional representation of the V information features.

(A) Represents the collected boundary voltage signals, (B) is their visualized image, and the variation in the waveform reflects the changes in voltage at different collection positions. (C) Is a two-dimensional matrix reconstructed according to Eq. (9), (D) is the matrix form of c after transposition, and (E and F) demonstrate the data differences between different columns and rows.

(9) Vb=[V3,42,1  V4,52,1⋯V15,162,1 V17,182,1 V18,192,1 ⋯  V32,172,1 V33,342,1 V34,352,1 ⋯ V48,332,1V4,53,2  V5,63,2 ⋯ V16,13,2 V18,193,2 V19,203,2 ⋯  V17,183,2 V34,353,2 V35,363,2 ⋯  V33,343,2       ⋱              ⋱V1,216,15 V2,316,15 ⋯  V13,1416,15 V31,3216,15 V32,1716,15 ⋯ V30,3116,15 V47,4816,15 V48,3316,15 ⋯ V46,4716,15V1,218,17 V2,318,17 ⋯  V16,118,17 V19,2018,17 V20,2118,17 ⋯ V31,3218,17 V33,3418,17 V34,3518,17 ⋯  V48,3318,17V2,319,18 V3,419,18 ⋯  V1,219,18 V20,2119,18 V21,2219,18 ⋯ V32,1719,18 V34,3519,18 V35,3619,18 ⋯ V33,3419,18        ⋱              ⋱V16,117,32 V1,217,32 ⋯ V15,1617,32 V18,1917,32 V19,2017,32 ⋯ V30,3117,32 V48,3317,32 V33,3417,32 ⋯  V47,4817,32V1,234,33 V2,334,33 ⋯ V16,134,33 V17,1834,33 V18,1934,33 ⋯ V32,1734,33 V35,3634,33 V36,3734,33 ⋯  V47,4834,33V2,335,34 V3,435,34 ⋯ V1,235,34 V18,1935,34 V19,2035,34 ⋯ V17,1835,34 V36,3735,34 V37,3835,34 ⋯  V48,1735,34        ⋱              ⋱V16,133,48 V1,233,48 ⋯  V15,1633,48 V32,1733,48 V17,1833,48 ⋯ V31,3233,48 V34,3533,48 V35,3633,48 ⋯  V46,4733,48]48×45

In Vout1,out2in1,in2, in1 and in2 represent the excitation electrodes, while out1 and out2 correspond to the measurement electrodes.

3D EIT reconstruction network

Motivated by Eqs. (3) and (4), this study aims to model the nonlinear relationship between the conductivity distribution and the boundary voltage values, thereby proposing a solution for the 3D EIT inverse problem. This objective is further elaborated upon in Eqs. (6) and (7).

(10) σℝ(x,y,z)=F(Vb,In,R(w,b))

(11) {ϕl={ν1,ν2,⋯,νL},L=2160,ϕl∈∂ΩVb≡ϕlIn={i1,i2,⋯,iN},N=48,In∈∂ΩR(w,b)={(w1,b1),(w2,b2),⋯,(wm,bm)}.

σR represents the three-dimensional field conductivity distribution, while F symbolizes the neural network imaging algorithm. Vb aligns with element ϕl and solely undergoes reconstruction according to Eq. (9). In denotes the boundary excitation current, and R(w,b) embodies the weight and bias parameters of the neuron.

The original data, represented as a one-dimensional voltage signal of 2,160 × 1, is transformed into a single-channel voltage data of 45 × 48 according to the rules stipulated by Vb, serving as the network input for reconstruction. The initial convolution layer is devised with one input channel and 64 output channels. To expedite the convolution calculation, the kernel size, stride, and padding are set to 7, 2, and 3, respectively. The data size post the initial convolution is 64 × 23 × 24. Following Maxpool, the data is fed into the ‘Bottleneck’, which houses two structure types—‘Bottleneck_1’ and ‘Bottleneck_2’. Both utilize the Relu nonlinear activation function for intermediate activation. Post average pooling, the data size is transformed to 2,048 × 1 × 1 and finally, 95,295 conductivity data is outputted via the fully connected layer (fc). To avoid excessive resource occupation and simultaneously ensuring the validity of the output data, a single layer fully connected mapping is employed for outputting the conductivity results after numerous experiments. This method aims for the fc output to adhere to the target conductivity trend as closely as possible, ultimately procuring an approximate value that fluctuates within a certain range. This embodies a regression analysis approach. Upon observing the reconstruction results, notable differences can be discerned between the reconstructed conductivity and the target conductivity. However, the error values of back propagation remain small and traditional methods of calculating the loss cannot effectively constrain the model to converge. To mitigate this, the error term is amplified in this article, enabling the convolutional model to identify larger error terms more easily and effectively converge the model. An overview of the method proposed in this article is depicted in Fig. 3.

Figure 3 Schematic representation of the 3D EIT reconstruction network.

The 3D EIT convolutional network includes two types of residual blocks, Bottleneck_1 and Bottleneck_2. The input to the network is a two-dimensional matrix reconstructed from boundary voltages, and the output is one-dimensional conductivity, as shown in (A), which visualizes the difference between the predicted conductivity and the true conductivity. Part (B) provides a more detailed description of the difference.

Materials and datasets

Experimental instrument

The open-source software package “Electrical Impedance Chromatography and Diffuse Optical Chromatography Reconstruction Software” (EIDORS 3.10 with Netgen 5.3) (Adler, 2019) was employed for the generation of simulated experimental data. The actual data were sourced from the EIT system, designed and developed by our research team (Li et al., 2022), as represented in Fig. 4.

Figure 4 Depiction of the EIT measurement system.

(A) Is the overall view of the hardware system, (B) is a view of the data acquisition interface, and (C) is the left view.

The 3D EIT boundary voltage acquisition used adjacent current excitation and adjacent voltage measurement, where one pair of adjacent electrodes was selected as the initial excitation end and the other two adjacent electrodes were selected as the measurement end. The system uses a three-layer electrode structure as shown in Fig. 5A. The excitation electrode uses cyclic excitation, while the measurement electrode performs periodic measurements in the order of Eq. (9). The conductivity distribution σR was calculated by FEM, and the voltage distribution ϕ(x,y,z) in the field was obtained from Eq. (2), and 2,160 boundary voltages were collected according to the principle of adjacent excitation-adjacent measurement. Finally, the conductivity values σ′(x,y,z) are reconstructed by the algorithm F. The basic principle is shown in Fig. 5.

Figure 5 Fundamentals of 3D EIT.

(A) Represents the excitation-measurement model, with its finite element model being (B). (C) Shows the simulated potential distribution, and (D) represents the waveforms of the measured boundary voltages, where No. 1–3 indicate waveforms measured at different electrode rings. (E) Depicts the predicted conductivity and the reconstructed conductivity distribution.

Datasets

Simulation data

In this article, a simulation dataset for training RFNetEIT was created using EIDORS 3.10; details of the production process can be found in the Supplemental Material. The dataset consists of absolute imaging data that capture the conductivity distribution. Voltage-conductivity samples are generated through finite element modeling simulation. Each sample comprises two one-dimensional vectors: the boundary measurement voltage vector and the true conductivity distribution vector. In the EIT system, which features a cylindrical arrangement of 16 × 3 electrodes as illustrated in Fig. 5, we employ adjacent excitation and adjacent measurements. A complete measurement cycle in this setup contains m = 2,160 voltage measurements.

Acrylic model: The Acrylic model simulation dataset comprises simulation data featuring various objects to be measured and different signal-to-noise ratios. A total of 10,000 samples are generated for each type of data and are randomly divided into training and validation sets at an 8:2 ratio. Additionally, 100 test samples are regenerated to provide a statistical characterization of the experimental results. In this setup, the measurement domain is divided into a fixed number of n = 95,295 cells using finite elements. The background medium is salt water, with a constant conductivity of 350μS/cm, which remains unaltered across frequencies. The objects to be measured in the simulation are spheres, cylinders, and cubes made of acrylic, with the conductivity of each object set at 0.0001 μS/cm. Besides conducting an ideal experiment, system noise is also simulated by adding Gaussian noise. In accordance with the conditions set by the signal detection limit, the measured signal needs to exceed the sum of the noise mean and three times the noise standard deviation, as described in Eq. (15). Table 1 presents the signal occupancy ratios required to meet the signal detection limit under various SNRs, and range 30dB≤SNR≤50dB. has been selected for the experiment. The details of the simulation environment are provided in Table 2.

Table 1 Proportion of signals adhering to the signal detection limit condition.

SNR	Number of data points	Percentage of signals meeting the limit	
20 dB	602	27.9%	
30 dB	1,088	50.4%	
40 dB	1,406	65.1%	
50 dB	1,558	72.1%	
60 dB	1,606	74.4%	
Note:

The first column, “SNR,” stands for signal-to-noise ratio, measured in decibels (dB); the second column, “Number of Data Points,” shows the count of voltage data points that meet the signal detection threshold; the third column, “Percentage of Signals Meeting the Limit,” indicates the percentage of signals that conform to the detection threshold.

Table 2 Simulation environment configuration.

Variables	Parameters	
Field dimensions	H: 8 cm, R: 5 cm	
Number of electrode layers	3	
Number of electrodes	16*3	
Number of finite element cells	95,295	
Background conductivity	350μS/cm	
Driven pattern	Adjacent	
Type of object to be measured	Sphere, Cylinder, Cube	
SNR	30 dB, 40 dB, 50 dB	
Conductivity of object to be measured	0.0001 μS/cm	
Note:

The table details the configuration of a simulation environment, specifying the dimensions and properties of the field, the number and arrangement of electrodes, the background medium’s conductivity, and the types and characteristics of objects being measured, under various signal-to-noise ratio conditions.

Maize ear model: In the maize ear simulation dataset, there are 20,000 samples, which include 16,000 training samples and 4,000 validation samples. The measurement domain comprises 106,881 FEM units. To simulate the moisture distribution within the maize ear, a gradient decaying conductivity distribution was established, emanating from the center to the edge of the model for the forward problem. This was achieved by setting the value of f(x) to decay exponentially according to the Euclidean distance between each computational unit and the center of both models, as described in Eq. (12). Here, x represents the distance from the center point, while s is a custom attenuation parameter. To more closely mirror real-world conditions, centroids were randomly selected along the X, Y, and Z axes. A sinusoidal function s(z) with small amplitude was employed to represent the variation of the diffusion scale in the Z direction, as outlined in Eq. (13), where z denotes the center point’s position on the Z-axis. Finally, the conductivity for each element was calculated using both the Euclidean distance and the diffusion scale factor in the Z direction, as illustrated in Eq. (14). In this equation, Dxy and DZ respectively represent the Euclidean distance from the center of each element in the plane to the specified center, and bs is a triad defining the fundamental decay rate in each direction.

(12) f(x)=e−x/s

(13) s(z)=1+asin(bz+c)

(14) Es=e(−Dxy×s(z)bs_1)+(−Dxy×s(z)bs_2)+(−DZbs_3)

(15) ν>μ(εnoise)+3×σ(εnoise).

Measurement data of acrylic models

In this experiment, a NaCl-based solution with a conductivity of 350μS/cm, and a fluctuation range of ±3% was used as the background medium. The test subjects included a sphere, a cylinder, a cube, and a combination of all three, as depicted in Fig. 6.

Figure 6 Different forms of acrylic models, (A) sphere, (B) cube, (C) cylinder, (D) 3 acrylic models.

(A–D) Represent different acrylic models, specifically a sphere, a cube, a cylinder, and three other acrylic models.

Measurement data of complete maize ears

Freshly picked maize cobs were selected for the experiment, and the measurement areas of interest included the husk, kernels, and cob. The electrical properties of biological tissues vary greatly across different frequency ranges (Cao et al., 2019). According to the impedance spectrum-based frequency selection method by Li et al. (2023), the magnitude of the impedance change in maize cobs decreases after 8 kHz, and the impedance hardly changes any further in the frequency range greater than 80 kHz. Therefore, in this article, we employ a simulated, noise-free, ideal boundary voltage as a benchmark to evaluate the quality of the measured signal at excitation frequencies within 80 kHz, based on the similarity between the global and local distribution characteristics of the boundary voltage data points.

Maximum mean discrepancy (MMD)

Distributional differences are measured by calculating the distance between the kernel matrices of the two sample sets in the feature space, as shown in Eq. (16).

(16) MMD=1n(n−1)∑i≠jk(xi,xj)+1m(m−1)∑i≠jk(yi,yj)−2nm∑i,jk(xi,yj)

where n is the size of the sample set X, m is the size of the sample set Y, xi and yi are the ith samples in the sample sets X and Y, respectively, and k(x,y) represents the value of the kernel function (typically a radial basis kernel function is employed, such as a Gaussian kernel function).

Kernel density estimation (KDE)

The difference between the distributions of the two sample sets X and Yis compared by calculating the integral squared difference (ISE) as shown in Eq. (17).

(17) {f^X(x)=1nXhX∑i=1nXK(x−xihX)f^Y(y)=1nYhY∑j=1nYK(y−yjhY)KDEISE(f^X,f^Y)=∫(f^X(x)−f^Y(y))2dx

where x and y denote data points, nX and nY denote the number of data points, K denotes the kernel function, and hX and hY denote the bandwidth parameters.

Local maximum mean discrepancy (LMMD)

It can be used to measure the average of the MMDs of the sample sets X and Y in different local windows, thus reflecting the local distributional differences. The LMMD is calculated as follows:

(18) LMMD(X,Y)=1N∑i=1N⁡MMD(X,Y)

where N is the number of windows.

k-nearest neighbor distribution complexity (KNNDC)

KNNDC quantifies the distributional complexity between two sample sets by calculating the average distance that each data point in sample set X needs to traverse to locate its k nearest neighbors within sample set Y. Similarly, the KNNDC denotes the average distance required by each data point in sample set Y to identify its k nearest neighbors. The formula is presented below:

(19) KNNDC(x,Y,k)=1k∑i=1kD(x,yi)

where yi is the i-th nearest neighbor of x in the sample set Y, and D(x,yi) is the distance (e.g., Euclidean distance) between x and yi.

Combining the results of the four indicators in Fig. 7, the similarity between the measured boundary voltage and the reference ideal voltage is high when the excitation frequency is in the range of 1–10 kHz. Therefore, the excitation frequencies selected for the experiment are 2,000, 4,000, 6,000 and 8,000 Hz.

Figure 7 Quality assessment of measured boundary voltage using (A) MMD, (B) KDE, (C) LMMD, (D) KNNDC.

(A–D) Describe the results of four evaluation metrics. (A–C) Show that based on the results, the metrics begin to stabilize after the frequency reaches 4,180 Hz. The results in (D) indicate that within the frequency range of 4,180 to 8,160 Hz, the KNNDC metric for the five samples remains at or near its minimum value.

Evaluation indicators

The algorithm’s performance is evaluated using root mean square error (RMSE), correlation coefficients (CC), structural similarity index (SSIM), and inverse problem solving time (IPST).

The RMSE is widely used to measure the deviation between predicted and true values:

(20) RMSE(P,A)=∑i=1N(Pi−Ai)2N.

CC is commonly used as a statistical measure to quantify the degree of correlation between two variables. In this article, it is used to evaluate the correlation between two images, as described by the following equation:

(21) CC(P,A)=σPiAiσPi2⋅σAi2.

SSIM is a measure of the degree of structural similarity between images:

(22) SSIM(P,A)=(2μPiμAi+c1)(2σPiAi+c2)(μPi2+μAi2+c1)(σPi2+σAi2+c2)

where N denotes the number of samples, P denotes the reconstructed image, and A denotes the actual image. The variables μ, σ2, and σPiAi represent the mean, variance, and covariance of Pi and Ai, respectively. Constants c1 and c2 are also involved in the equation.

In 3D electrical impedance tomography, solving the inverse problem can be time-consuming due to the large volume of reconstructed data. Therefore, in this article, the inverse problem solving time (IPST) is used as a measure to evaluate the computational efficiency of the algorithm.

Computational platform and experimental details

The computer used for the experiments had an Intel Core i5-10400F 2.90 GHz hexa-core CPU, an NVIDIA GeForce RTX 3060 (12 GB) GPU, and 32 GB of RAM. Model simulation was performed using MATLAB R2022b on a Windows 10 OS. Network training was performed in the PyTorch framework, with an integrated development environment (IDE) of PyCharm and a compiled language of Python. Training data generation and imaging were based on the open platform EIDORS 3.10.

The model training process does not use a pre-trained model. The epoch is set to 50, the batch size is 32, and the loss function is mean square error loss (MSE Loss). The adaptive moment estimation (Adam) optimizer is used, along with a dynamic learning rate adjustment strategy. The initial learning rate is set to 0.0001.

Results

Extensive experimental results are presented in this section and compared with results from classical absolute reconstruction algorithms (GN solver, constrained Gauss-Newton solver (cGN)) and widely used end-to-end models (e.g., UNet).

Simulation experiment

The algorithm’s reconstruction performance is evaluated through experiments on simulated single and multi-object scenarios. The results are compared with the classical GN and cGN algorithms, as well as the widely used UNet. The ability of the algorithm to accurately reconstruct the shape and spatial position of the measured objects is assessed qualitatively through 3D visualization. Additionally, the reconstruction accuracy is assessed quantitatively using metrics such as RMSE, CC, SSIM, and IPST. The experiments demonstrate that the proposed method effectively reconstructs the size and position of both single and multiple targets. RFNetEIT achieves optimal reconstruction quality in terms of shape constraints and position localization for each measured object. The evaluation metrics, including RMSE, CC, SSIM, and IPST, indicate excellent results, further validating the effectiveness of the proposed method. Table 3 presents a comparison of the reconstruction results under four different levels of noise levels. The results show notable differences in the quality of reconstruction between the conventional algorithms (GN and cGN) and the deep learning algorithms (UNet and Ours) for Objects 1 to 4 under different noise levels. Particularly, the discrepancy in reconstruction results is most significant between the 30 dB noise condition and the noiseless ideal condition for the same object. Taking Object 1 (sphere) as an example, the three-dimensional slices obtained from the GN and exhibit evident differences in reconstruction, with noticeable changes in morphology as noise increases.

Table 3 Reconstruction results at different SNRs.

		Object 1	Object 2	Object 3	Object 4		
							
None	GN						
cGN						
UNet						
Ours						
50 dB	GN						
cGN						
UNet						
Ours						
40 dB	GN						
cGN						
UNet						
Ours						
30 dB	GN						
	cGN						
	UNet						
Ours						
Note:

Objects 1–4 represent different simulation models, namely, sphere, cube, cylinder, and three acrylic models. ‘None,’ ‘50 dB,’ ‘40 dB,’ and ‘30 dB’ denote the signal-to-noise ratios of the simulation experiments. At each signal-to-noise ratio, qualitative comparisons are made between GN, cGN, UNet, and our method, including the reconstruction results for the four simulation models from Object 1 to 4, respectively. The imaging results for each model are presented through three-dimensional views and three-dimensional sliced views, with conductivity levels displayed using colorbar.

Specifically, under the 30 dB signal-to-noise ratio (S/N) condition, both GN and cGN can roughly reconstruct the morphology of Object 1 (sphere) and Object 2 (cylinder). However, when it comes to Object 3 (cube) and Object 4 (multiple objects), the GN algorithm shows vague conductivity changes in the field, while cGN is limited in reconstructing them. On the other hand, UNet can capture changes in the shape and number of objects, but it struggles with the accurate reconstruction of the edges of Object 3 and Object 4. The proposed method in this article demonstrates superior reconstruction quality for objects with varying shapes and quantities. The reconstructed objects exhibit uniform conductivity distributions with minimal artifacts, as observed from the slice results. Among the four sets of experiments, Object 4 presents the greatest reconstruction challenge. The GN can only approximate the conductivity range, while the cGN struggles to effectively reconstruct the visualized conductivity. The UNet incorrectly reconstructs four cylinders in the noiseless and SNR = 40 dB experiments, but produces clearer conductivity distributions at SNR = 50 and 30 dB, albeit with poor reconstruction quality for the spheres. In contrast, the method proposed in this article achieves high-quality reconstruction for objects with diverse shapes.

Table 4 shows the IPST comparison results of the four methods in the Object 1–4 experiments. The computation time range of the inverse problem for traditional methods such as GN and cGN is roughly 60–200 s, which mainly depends on the CPU and memory usage. While the time required by UNet and RFNetEIT is within 0.1 s, although the computation time of RFNetEIT is slightly higher than that of UNet, the results of the metrics such as RMSE, CC, and SSIM are better. In Table 5, the RMSE for RFNetEIT is 0.0029 with a 95% confidence interval of (0.00274, 0.00306), while for UNet it is 0.0052 with a confidence interval of (0.00508, 0.00532). For CC, RFNetEIT yields a mean of 0.9504 with a confidence interval of (0.9394, 0.9614), compared to UNet’s 0.846 (0.836, 0.856). The SSIM value for RFNetEIT is 0.9923 with a confidence interval of (0.9909, 0.9937), whereas UNet presents 0.9769 (0.9757, 0.9781). As the SNR decreases, significant differences are presented between RFNetEIT and the comparison methods (experimental results for RMSE, CC, and SSIM are in the Supplemental Material). Among these, the contrast is most pronounced at 30 dB. Using the comparison experiment for Object 2 as an example, the range of RMSE for the method presented in this article is primarily between 0.002 and 0.003, the range of CC is approximately between 0.93 and 0.98, and the SSIM is mostly 0.99 or higher. Based on data from 100 samples, the fluctuations in the method presented in this article are minimized, indicating strong robustness. Additionally, the confidence intervals for the Object 4 experiments are shown in Table 6. Specifically, the confidence intervals for this article’s method are [0.274,0.306]*0.01 for RMSE, [0.9394, 0.9614] for CC, and [0.9909, 0.9937] for SSIM, respectively.

Table 4 Comparison of solution times for inverse problems.

		Unit: second (S)	
		GN	cGN	UNet	Ours	
Object 1	None	87.4506 ± 16.8702	64.6634 ± 4.4875	0.0138 ± 0.0818	0.0386 ± 0.2584	
50 dB	80.2264 ± 18.2462	120.1662 ± 2.0118	0.0055 ± 0.0006	0.0128 ± 0.0008	
40 dB	89.2327 ± 17.3967	64.9144 ± 1.8018	0.0138 ± 0.0815	0.0620 ± 0.4900	
30 dB	79.0027 ± 20.1494	118.4615 ± 1.4094	0.0296 ± 0.2421	0.0231 ± 0.1022	
Object 2	None	125.2902 ± 11.2747	63.7808 ± 2.9339	0.0305 ± 0.2470	0.0958 ± 0.8253	
50 dB	127.4050 ± 11.5543	121.8856 ± 2.5243	0.0059 ± 0.0017	0.0123 ± 0.0012	
40 dB	127.6764 ± 10.1286	68.9898 ± 5.4898	0.0140 ± 0.0841	0.0387 ± 0.2673	
30 dB	123.8375 ± 12.4858	120.1919 ± 2.7749	0.0055 ± 0.0006	0.0128 ± 0.0009	
Object 3	None	118.0406 ± 13.9895	75.6850 ± 1.0838	0.0145 ± 0.0873	0.0514 ± 0.3992	
50 dB	123.8820 ± 15.6049	118.2292 ± 0.9340	0.0054 ± 0.0005	0.0129 ± 0.0010	
40 dB	116.7123 ± 14.1016	76.4561 ± 1.0054	0.0142 ± 0.0854	0.0409 ± 0.2918	
30 dB	121.0006 ± 16.8621	118.5108 ± 1.3474	0.0053 ± 0.0006	0.0127 ± 0.0010	
Object 4	None	212.3173 ± 15.9137	119.9486 ± 1.3198	0.0144 ± 0.0884	0.0188 ± 0.0822	
50 dB	126.8463 ± 9.1561	116.4603 ± 0.5347	0.0055 ± 0.0006	0.0126 ± 0.0010	
40 dB	209.2098 ± 17.4149	118.1869 ± 0.6031	0.0139 ± 0.0836	0.0190 ± 0.0860	
30 dB	126.9345 ± 8.1161	116.6223 ± 1.3403	0.0055 ± 0.0005	0.0124 ± 0.0010	
Note:

GN, cGN, UNet, and ours are the methods to be compared. Objects 1–4 represent four types of test objects, namely sphere, cube, cylinder, and three acrylic models. None, 50, 40, and 30 dB represent the signal-to-noise ratios in simulation experiments. ‘87.4506 ± 16.8702’ indicates that ‘87.4506’ is the mean time consumed in 100 sample inverse problem-solving experiments, and ‘16.8702’ is the standard deviation.

Table 5 Evaluation indicators for the 30 dB SNR simulation experiment.

	Object 1	Object 2	Object 3	Object 4	
30 dB	RMSE					
CC					
SSIM					
Note:

Objects 1–4 represent experiments with four types of test objects, and the displayed results are from experiments conducted under a 30 dB signal-to-noise ratio condition. RMSE, CC, and SSIM represent three evaluation metrics. The results of the comparative experiments are represented by curves in four different colors. Black represents GN, green corresponds to cGN, blue represents UNet, and red represents our proposed method, RFNetEIT. The number of experimental samples is 100.

Table 6 Confidence intervals (95% confidence level) for object 4 on each indicator at 30 dB.

	RMSE	CC	SSIM	
GN	[0.718, 0.742] *0.01	[0.6597, 0.6798]	[0.9528, 0.9556]	
cGN	[0.853, 0.915] *0.01	[0.5945, 0.6145]	[0.9356, 0.9406]	
UNet	[0.508, 0.532] *0.01	[0.836, 0.856]	[0.9757, 0.9781]	
Ours	[0.274, 0.306] *0.01	[0.9394, 0.9614]	[0.9909, 0.9937]	
Note:

GN, cGN, UNet, and Ours represent the comparison methods, while RMSE, CC, and SSIM are the evaluation metrics. [0.718, 0.742] *0.01 indicates that the confidence interval for the GN method on RMSE is [0.00718, 0.00742]. In the table, the subscript *0.01 is used for the convenience of displaying the results. The bolded text indicates the results of this study.

Table 6 presents the confidence intervals for the different methods in the Object 4 experiment. For the proposed method, the confidence intervals are [0.274, 0.306]*0.01 for RMSE, [0.9394, 0.9614] for CC, and [0.9909, 0.9937] for SSIM. These intervals are significantly better than those of UNet, which are [0.508, 0.532]*0.01 for RMSE, [0.836, 0.856] for CC, and [0.9757, 0.9781] for SSIM. For the confidence intervals of each indicator at different signal-to-noise ratios, please refer to the Supplemental Material.

Experiments with acrylic models

A solution of 350μS/cm and an acrylic model serve as the experimental objects. A training model, augmented with Gaussian noise based on numerical simulation experiments, is employed as a priori information to reconstruct the conductivity distribution of the acrylic in the measured experiments. Comparative experiments are conducted using the methodology described in “Simulation Experiment”. Given that it is not feasible to quantify the experimental objects and test fields, the quantitative metrics employed in the simulation experiments are not directly applicable to these real-world tests. Accordingly, this section evaluates the results of the experiments through qualitative analysis. The specific objects tested are listed in Table 7.

Table 7 Image reconstruction results based on measured data.

Excitation frequencies	Method	Object 1	Object 2	Object 3	Object 4		
					
1,000 Hz	GN						
cGN						
UNet						
Ours						
10,000 Hz	GN						
cGN						
UNet						
Ours						
50,000 Hz	GN						
cGN						
UNet						
Ours						
Note:

The excitation frequencies chosen for the actual measurement experiments were 1, 10 and 50 KHz. The comparison methods at each frequency are GN, cGN, UNet, and Ours. The experimental subjects are Objects 1–4. The imaging results of each method at different frequencies and on different objects are displayed through three-dimensional views and three-dimensional slice views, and the conductivity levels are shown using a colorbar.

Experiments were conducted using different stimulation frequencies (1, 10, and 50 kHz) with acrylic models of various shapes (sphere, cylinder, rectangle, and multi-target) placed in the container as reconstruction targets. The results of all reconstructions are shown in Table 7. The GN is able to capture coarse conductivity variations in the measurement area, although the imaging quality is influenced by the excitation frequency. However, the cGN has difficulty in accurately representing conductivity variations at all excitation frequencies. On the other hand, the UNet can reconstruct the image, but the position and shape of the reconstructed object are biased. For single target objects (Object 1–Object 3), the object reconstructed using RFNetEIT exhibits the best position and shape. However, for Object 4, both the GN and cGN struggle to reconstruct clearer targets due to the complexity of the target and the presence of hardware system noise, while the UNet produces incorrect reconstruction results. Three slices of each reconstruction result are shown in Table 7. A comparison of the slices reveals that the GN is not suited for reconstruct a uniform conductivity distribution, the UNet generates a significant artifact in the target conductivity, while RFNetEIT achieves a more uniform conductivity distribution in 3D space.

Reconstructing the electrical conductivity distribution of maize ears

In the experiments, currents were excited at frequencies ranging from 2 to 8 kHz. Three layers of electrodes were mounted on the maize ear. The electrode positions remained stationary during measurements, as depicted in Fig. 8A. “Simulation Experiment” shows that the UNet lacks robustness in 3D imaging, while “Experiments with Acrylic Models” discusses how the cGN is less effective for accurately reconstructing the 3D conductivity distribution. To evaluate the performance of our proposed method, we combined these findings with current research on the 3D conductivity reconstruction of maize ears. The comparative algorithms we selected include the interpolation imaging method by Li et al. (2022) and the GN algorithm.

Figure 8 Reconstructed conductivity distribution using measured voltages from maize ear.

(A) Is the 3DEIT measurement system for maize ear, (B) shows a schematic of the structure of the area to be measured, and (C) is the transverse and longitudinal sections of the maize ear domain to be measured. After the boundary voltage was acquired by system A, conductivity reconstruction was performed using the GN method shown in (D) and the method in this article, and differential imaging was performed using the interpolation method shown in (E). The conductivity reconstruction results of the two methods at 2, 4, 6, and 8 KHz are shown in (D). The conductivity reconstruction results of the cross-section in layers ①–③ are shown in (E).

Figure 7A displays the complete maize ear measurement apparatus. Since the maize ear has a three-layer structure, the main part of the ear is selected as the imaging target, as shown in Fig. 8B. Figure 8C represents the conductivity model using transverse and longitudinal slices to better illustrate the distribution. Figure 8D contrasts the absolute reconstruction results, and Fig. 8E displays the reconstructed frequency difference image obtained through interpolation. In Li’s experiment (Li et al., 2022), three layers of electrodes were also used, but the electrodes on different layers operated independently. A separate set of voltage data was excited and measured for each layer, resulting in a total of 208 × 3 voltage measurements. Compared to the absolute reconstruction results in Fig. 8D, Li’s method only demonstrates visual three-dimensional imaging. The conductivity between the layers of electrodes is estimated using interpolation methods, as the true value cannot be calculated using EIT theory. In Fig. 8D, the GN can reconstruct variations in conductivity distribution, but the differences are significant at varying excitation frequencies. In contrast, the method proposed in this article provides the best reconstruction results, and these results are stable across different frequencies. Figure 9 displays the conductivity reconstruction results for five maize ears using a uniform colorbar, revealing differences in conductivity distribution among different maize ears.

Figure 9 Reconstruction results of conductivity distribution of five maize ears.

Samples 1–5 are fresh maize ears, measured using the same equipment. The second row shows the conductivity reconstruction results for the corresponding maize ears. Different conductivity levels are shown using the colorbar.

Discussion

The electrical impedance distribution has the potential to indirectly reflect the physiological state of the maize ear. However, existing EIT imaging studies fall short in completely reconstructing the electrical conductivity in the 3D space of the bract, seed, and cob, which hinders an accurate representation of the true condition of the maize ear. To address this challenge, this article proposes a nonlinear mapping model based on ResNet. By designing a dedicated single-channel CNN and implementing data reconstruction rules, the model's stability is significantly improved, thereby mitigating the underdetermined and nonlinearity issues encountered in solving the inverse EIT problem. Additionally, the ill-conditioned problem is alleviated by training an a priori model that incorporates noise. The proposed method demonstrates excellent shape constraints and achieves accurate conductivity reconstruction in both simulation and real experiments. It proves effective in reconstructing the conductivity distribution of husk, kernels, and cob.

Comparison of reconstruction quality in numerical simulation experiments

GN and cGN algorithms struggle to accurately reconstruct the morphology and position of the simulated object due to challenges in determining suitable parameter values for constructing Jacobi matrices with good generalization and robustness. Additionally, the impact of system noise is a significant factor contributing to the poor or even non-imaging quality. The UNet network’s performance is not adequate to achieve a stable model due to the high number of reconstructed 3D conductivities and the complexity of the training dataset, resulting in significant fluctuations in the imaging results. In contrast, the method proposed in this article, incorporating a residual structure, exhibits improved feature extraction capability and network stability. As a result, it can more accurately approximate the distribution pattern of the real conductivity of the object under measurement, leading to superior imaging quality and evaluation metrics compared to the comparison method.

The reconstruction difficulty is in the order of: multi-target > cube > cylinder > sphere. The reconstruction difficulty of multi-target is the highest, followed by rectangular. One possible reason is that the rectangular structure is more complex compared to the sphere and cylinder simulation models, with a surface shape that is rectangular and the presence of vertical transitions on adjacent surfaces. Moreover, the field containing multiple objects to be measured exhibits a greater variation in conductivity, resulting in the highest level of reconstruction difficulty.

Reconstruction quality comparison of acrylic measurement experiments

The limited ability of GN and cGN to accurately reconstruct the conductivity distribution of the acrylic model can be attributed to the influence of environmental noise and hardware system noise. The difficulties faced by cGN in imaging are primarily due to the higher impact of such noise compared to the Gaussian noise added in the simulation experiments, resulting in decreased stability compared to the GN algorithm. Furthermore, variations in the excitation frequency can lead to changes in the boundary voltage, further contributing to differences in the reconstruction results.

For single-target reconstruction, the experimental and simulation results of this article are close to each other, and the deviation from the shape size and spatial position of the real object to be measured is minimal. However, in multi-target reconstruction, the accuracy of the reconstructed shape of the measured object needs further improvement due to the influence of system noise and model complexity. UNet is prone to overfitting due to the limited feature extraction capability of the model, and the training process is prone to produce incorrect reconstruction results, which is more obvious in multi-target reconstruction.

Comparison of reconstructed quality of maize ear experiments

Impedance distribution within the maize ear correlates with its moisture content. Factors such as humidity, temperature, and maize maturity significantly influence impedance. Typically, regions with higher moisture content exhibit lower impedance. Thus, internal impedance variations in the maize ear could serve as an indirect method to evaluate moisture levels. However, challenges arise due to the non-destructive nature of the acquisition devices and impedance differences across electrodes at various maize ear positions, complicating conductivity reconstruction.

The study by Li et al. (2022) used three layers of independently operating electrodes to generate 2D EIT images, which were then interpolated by an interpolation algorithm to visualize 3D conductivity distributions, but this method fills in the missing conductivity values by calculating linear interpolation between the known points, and is not the true conductivity distribution of the maize ear. While the imaging effect of the GN is greatly affected by the excitation frequency, it is a challenge to determine the optimal excitation frequency in practical applications.

In contrast, the absolute imaging results of the method in this article can demonstrate relatively complete 3D structural information. The reconstruction results from RFNetEIT can clearly distinguish between husk, kernels, and cob. Furthermore, the conductivity distribution within each part shows gradient changes, which aligns with the actual conditions. However, due to the low signal-to-noise ratio in the actual boundary voltage measurements of the maize ear and the intricate physiological attributes and structure of the maize ear, obtaining authentic training samples becomes a challenge. This has led to the current study only being able to observe the 3D conductivity distribution within the maize ear without accurately reconstructing the conductivity values for specific sections.

Conclusions

This study presents a 3D EIT absolute imaging method, grounded in a deep convolutional model, devised to reconstruct the conductivity distribution across the tri-layer structure (husk, kernels, and cob) of a complete maize ear. By combining the strengths of traditional residual neural networks and error-constrained terms for conductivity prediction, we established a nonlinear mapping model of boundary voltage and 3D conductivity spatial distribution. The boundary voltage information was transformed into a data feature map that includes spatial location information, increasing correlation across different dimensions and reducing the risk of model overfitting through a dynamic learning rate adjustment strategy.

The results from both simulation and real-world experiments demonstrated that the proposed method outperformed the comparison method in terms of imaging quality and noise robustness. These findings validate the feasibility of solving the 3D EIT image reconstruction problem using a data regression analysis approach. Moreover, the method exhibits great potential for extension to other application scenarios involving 3D imaging tasks.

Owing to the in-situ spatial-temporal moisture pattern of maize ears and its fluctuation throughout the growth phase, it plays a crucial role in breeding varieties that possess favorable agronomic traits. Therefore, the primary focus for future endeavors is to perform three-dimensional imaging based on the actual measurement of the electrical impedance distribution of the maize ear in vivo and in situ.

Supplemental Information

Supplemental Information 1 Code.

Supplemental Information 2 Supplementary Material.

Additional Information and Declarations

Competing Interests

Author Contributions

Data Availability

The authors declare that they have no competing interests.

Hai-Ying Zheng conceived and designed the experiments, performed the experiments, analyzed the data, performed the computation work, prepared figures and/or tables, authored or reviewed drafts of the article, and approved the final draft.

Yang Li performed the experiments, performed the computation work, authored or reviewed drafts of the article, and approved the final draft.

Nan Wang analyzed the data, performed the computation work, authored or reviewed drafts of the article, and approved the final draft.

Yang Xiang performed the experiments, performed the computation work, authored or reviewed drafts of the article, and approved the final draft.

Jin-Hang Liu analyzed the data, performed the computation work, prepared figures and/or tables, authored or reviewed drafts of the article, and approved the final draft.

Liu-Deng Zhang performed the experiments, authored or reviewed drafts of the article, and approved the final draft.

Lan Huang conceived and designed the experiments, authored or reviewed drafts of the article, and approved the final draft.

Zhong-Yi Wang conceived and designed the experiments, authored or reviewed drafts of the article, and approved the final draft.

The following information was supplied regarding data availability:

The code is available in the Supplemental File.

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
