# Peer review of "A novel framework for three-dimensional electrical impedance tomography reconstruction of maize ear via feature reconfiguration and residual networks"

_PeerJ Computer Science, doi:10.7717/peerj-cs.1944_

## Round 0.1 · original submission · Minor Revisions

Dear Authors,
Your paper has been revised. It has been considered interesting and worth to be considered for publication. However, it needs minor revisions. In particular, it has some typos throughout its sections. Furthermore, the section references also show some typos. I advise you to revise the manuscript carefully, following the indications of the reviewers in a way that will fix them.

**Language Note:** The Academic Editor has identified that the English language must be improved. PeerJ can provide language editing services - please contact us at [email protected] for pricing (be sure to provide your manuscript number and title). Alternatively, you should make your own arrangements to improve the language quality and provide details in your response letter. – PeerJ Staff

Reviewer 1 ·

Basic reporting

Overview: The topic of reconstruct the 3D electrical conductivity distribution of plants is very interesting and an advancement of the 2D measures commonly used. The use of the novel computational framework is exciting and will stimulate other researchers in the field to likely consider such measures and use of addressing the electrical nature in plants and the surroundings. Others in the research community will enjoy reading this study and learning about the approach used. The study herein is well documented and clear to a reader. The manuscript was a pleasure to read, and the results are presented well. This study will likely engage others to consider such measures.

It was nice to read specifically the aspects this paper addresses as : “The main contributions of this paper are as follows:”
The Acrylic Model to help work out measurements is a unique approach and a good model for addressing the impacts in distribution of the signals.

Experimental design

Minor comments:
1. The reference list is not correct
Line 67: “67 can be reflected by the electrical impedance distribution (Zhao et al., 2015; Li et al., 2022).”
There is no citation to Li et al., 2022 listed
Other references are not complete such as what year is this ?
Line 829: Zhao Pengfei, Zhang Hanlin, Zhao Dongjie, Wang Zhijie, Fan Lifeng, Huang Lan, Ma Qin, Wang Zhongyi. Rapid on-line non-destructive detection of the moisture content of corn ear by bioelectrical impedance spectroscopy. Biol Eng 8
Please double check all citations in the text with the reference list and the format of the reference list with dates and consistent format.
There are a few aspects which do not apers to be clear to a reader.
Perhaps everything is defined in previous publications but a few clarifying questions if the authors don’t mind and to make sure such matters are described in the paper or in referenced in previous papers.

Of the 48 sets of excitation electrodes and 45 sets of measurement electrodes, it is a bit confusing on which electrodes are being excited and when? Is it one at a time or all at once?

To make the paper more accessible, there are several terms which could be defined or explained:

line 150: What does "stacking" layers actually do, or look like computationally? - this references another paper so it is probably answered there, but good to define.

line 177: what is "overfitting"?

line 297: "that" is not needed.

Validity of the findings

It is assumed the math is correct but it was not assessed in this review

Additional comments

None

Reviewer 2 ·

Basic reporting

Figures of the results should be improved and better explained. Each figure must be associated with a caption, describing how to read them. This also applies for example to table 7, where all the images are reported but it is not described.

Experimental design

no comment

Validity of the findings

No comment

·

Basic reporting

This is an overall well written paper, which seems to hide quite a lot of work and research behind it. The subject is up-to-date and at the continually developing engineering field.

Experimental design

The system architecture is sufficiently explained with understandable way. The signal figures and the system block diagram also assist to that result.
The article clearly defines the research question.
Methods described with sufficient information.
From the supplementary Information, it is fully explained the process of creating a dataset.
The simulated datasets were generated using EIDORS 3.10 in conjunction with Netgen 5.3.

Validity of the findings

The article includes software validation and verification. Matlab-codes are well written.
The data on which the conclusions are based provided.
Conclusions are well stated, linked to original research question & limited to supporting results.
Extensive experimental results are presented and compared with results from other algorithms and methods, such as classical absolute reconstruction algorithms (Gauss-Newton solver (GN), constrained Gauss Newton solver (cGN)) and widely used end-to-end models (e.g. UNet).
The conclusions are appropriately stated and connected to the original question investigated.

Additional comments

The proposed method exhibits great potential for extension to other application scenarios. An overall well written paper.

---

## Round 0.2 · accepted · Accept

Dear Authors,

Your paper has been revised. It has been accepted for publication in its present form. Congratulations on the acceptance of your manuscript, and thanks for your interest in submitting your work to PeerJ Computer Science.

Reviewer 1 ·

Basic reporting

The authors have made all the suggested changes from the 1st review.
The paper is nicely presented.

Experimental design

The methods are clear now that the authors have explained them in more detail.

Validity of the findings

The data appears to well presented and analyzed.

Additional comments

None.
Other researchers will enjoy reading this paper and building experiments on these reports.

Reviewer 2 ·

Basic reporting

Previous comments on the figures have been addressed by the authors, Now the figures are better described in the paper.

Experimental design

no comment

Validity of the findings

no comment